# Inflammatory monocytes promote pre-engraftment syndrome and tocilizumab can therapeutically limit pathology in patients

Linlin Jin[1,2,3], Zimin Sun [1✉], Huilan Liu[1], Xiaoyu Zhu[1], Yonggang Zhou[2,3], Binqing Fu[2,3], Xiaohu Zheng[2,3], Kaidi Song[1], Baolin Tang[1], Yun Wu[1], Jiang Zhu [1], Rui Sun [2,3], Zhigang Tian [1,2,3✉] & Haiming Wei [1,2,3✉]

Unrelated cord blood transplantation (UCBT) is an effective treatment for hematopoietic disorders. However, this attractive approach is frequently accompanied by pre-engraftment syndrome (PES), severe cases of PES are associated with enhanced mortality and morbidity, but the pathogenesis of PES remains unclear. Here we show that GM-CSF produced by cord blood-derived inflammatory monocytes drives PES pathology, and that monocytes are the main source of IL-6 during PES. Further, we report the outcome of a single arm, single-center clinical study of tocilizumab in the treatment of steroid-refractory severe PES patients (www.chictr.org.cn ChiCTR1800015472). The study met the primary outcome measure since none of the patients was nonrelapse death during the 100 days follow-up. The study also met key secondary outcomes measures of neutrophil engraftment and hematopoiesis. These findings offer a therapeutic strategy with which to tackle PES and improve nonrelapse mortality.

[1] Department of Hematology, the First Affiliated Hospital of USTC, Division of Life Sciences and Medicine, University of Science and Technology of China, Hefei, China. [2] Hefei National Laboratory for Physical Sciences at Microscale, the CAS Key Laboratory of Innate Immunity and Chronic Disease, Division of Life Sciences and Medicine, University of Science and Technology of China, Hefei, China. [3] Institute of Immunology, University of Science and Technology of China, Hefei, China. ✉email: zmsun@ustc.edu.cn; tzg@ustc.edu.cn; ustcwhm@ustc.edu.cn

Hematopoietic stem cell transplantation (HSCT) is an efficient therapy for patients with hematological malignancies (leukemia, myeloma, and lymphomas) and other hematological disorders (myelodysplasia and aplastic anemia)[1–4]. However, given the extensive range of human leukocyte antigen (HLA) polymorphisms, and the small size of modern families, the majority of patients who are in need of HSCT do not have a HLA-matched donor[5–8]. HLA–haploidentical transplantation has spread rapidly worldwide, and cord blood (CB) is also a good alternative source of hematopoietic stem cells. CB has many advantages as a stem cell source. For example, CB is more permissive of HLA disparity due to its lower immunogenicity[9]. CB can also be made available immediately when emergent HSCT is required[10]. Crucially, procuring CB is very easy and is devoid of any risk for the newborn infant.

There is a growing body of evidence to indicate that haploidentical donors and CB are promising options for patients who lack a related and a HLA-matched unrelated donor[11–14]. Although hematopoietic recovery is delayed, the incidence of acute graft-versus-host disease (aGVHD) and chronic GVHD (cGVHD), is lower in UCBT recipients[12,15,16]. Moreover, the incidence of relapse is lower in UCBT compared with HSCT with a HLA-matched or HLA-mismatched unrelated donor, indicating that CB may have a stronger graft versus leukemia (GVL) than bone marrow and peripheral blood stem cells (PBSC)[2,10]. Unfortunately, UCBT is usually accompanied by the emergence of unique early immune reactions that develop prior to neutrophil engraftment[17]. Lee et al. referred to these early immune reactions as "pre-engraftment syndrome" (PES)[18].

PES is common following UCBT and is characterized by noninfectious high-grade fever, skin rash, diarrhea and other clinical findings, including pulmonary infiltrates or body weight gain[19,20]. PES occurs within the first few days after UCBT; the onset of symptoms typically occurs 5 days to occasionally 11 days after the infusion of cord blood. The incidence of PES after UCBT has been reported ranging from 20% to 86.8%[17–26]. However, the only available treatment for PES is corticosteroids. In contrast, some severe PES patients are steroid-refractory. Since the success of UCBT is limited by mortality associated with severe PES, it is imperative to develop methods with which to control such toxicity and thus reduce mortality.

Unfortunately, however, the pathogenesis of PES remains unclear. PES has also been reported in patients who have never achieved engraftment[17], suggesting that PES may be a response to the infused cord blood. Plasma C-reactive protein (CRP) levels are known to be slightly elevated at the onset of PES. Since CRP is a nonspecific marker of inflammation, these findings may indicate that inflammation plays a pivotal role in PES. Many studies have shown that monocytes are the predominant cell type found in alveolar lavage fluid from patients with PES. It has also become clear that circulating monocyte subsets are a heterogeneous population, and play an important role in tissue repair, homeostasis, and inflammation[27]. It is generally considered that parturition is associated with an inflammatory response, and that monocytes are first recruited to the myometrium and then contribute to the physiological inflammation associated with labor. Monocytes are activated at the end of gestation, this is followed by the upregulation of various proinflammatory cytokines, such as IL-6 and TNF-α. Previous research has also shown that IL-6 can induce the secretion of oxytocin, and expression of the oxytocin receptor, in the myometrium in order to stimulate uterine contractions. Furthermore, it has been reported that the proportion of CD14+ cells is significantly higher in intervillous blood compared with peripheral blood[28]. Serum levels of GM-CSF, IL-6, tumor necrosis factor-α (TNF-α), G-CSF, and monocyte chemoattractant protein 1 (MCP-1) are all known to be high in cord blood and are highly correlated with each other[29]. GM-CSF is a hematopoietic growth factor with an emerging role across a range of inflammatory disorders. Monocytes are known to be sensitive to GM-CSF. However, the expression profile of GM-CSF, and whether monocytes derived from cord blood are responsible for the inflammatory response seen during PES, has yet to be investigated.

In this work, we comprehensively study monocytes from peripheral blood stem cells and cord blood. We provide evidence that monocytes derived from cord blood have inflammatory characteristics. Furthermore, monocytes are involved in the pathogenesis of PES and represent the main source of IL-6 during PES. Importantly, our results offer a therapeutic strategy with which to tackle steroid-refractory PES patients.

## Results

**UCBT predisposes patients to PES.** Since the year 2000, a total of 1545 hematopoietic stem cell transplantations have now been performed for the treatment of high-risk hematological disorders at the First Affiliated Hospital of the University of Science and Technology of China: 21 received bone marrow, 198 received bone marrow and PBSCs, 385 received PBSCs, and 941 received unrelated CB. The number of umbilical cord blood transplants clearly increased over the years (Fig. 1a). We compared the outcomes of 601 consecutive patients with hematological malignancy, or myelodysplastic syndrome, who received myeloablative conditioning, and then underwent HSCT; 439 patients received UCBT and 162 received a peripheral blood stem cell transplantation (PBSCT). Of the patients who underwent UCBT, 335 out of 439 developed PES, the symptoms of which onset typically occurs >7 days before neutrophil engraftment after UCBT. In our cohort of patients, the incidence of PES was 76.3% (95% confidence interval [CI], 72–80) in cord blood recipients. In stark contrast, the incidence of PES was 2.5% (95% [CI], 0.01–4.8) in patients who had undergone PBSCT (Fig. 1b).

In order to manage PES better, we developed a system with which to grade the severity of PES in individual patients and guide the treatment of PES (Supplementary Table 1). To further investigate the effect of PES on clinical outcomes, we compared outcomes in patients with different PES grades. Indeed, grade-3 PES patients experienced a significantly higher cumulative incidence of grade II–IV, III–IV aGVHD at day 100 (77.8%, 95% [CI], 28.1–95.1; 66.7%, 95% [CI], 23.5–89.3) and higher nonrelapse mortality (NRM) at 3 years (88.9%, 95% [CI], 21.6–99.1), resulting in lower 3-year survival rate (11.1%, 95% [CI], 0.6–38.8) and lower leukemia-free survival at 3 years (11.1%, 95% [CI], 0.6–38.8) compared with mild grade-0 PES patients (Fig. 1c–f, Supplementary Fig. 1). Grade-3 PES is the main factor responsible for early transplant-related mortality and represents a key deterrent to successful UCBT. To maximize the chances of therapeutic benefit, and minimize the risk of life-threatening complications of UCBT, there is an urgent need to delineate the mechanisms underlying PES and improve medical therapies.

**The number of monocytes was significantly increased in patients with PES.** Despite the fact that PES has been widely reported, very little is known of its underlying physiopathology. To determine whether CB grafts contribute to the pathogenesis of PES, we analyzed grafts between PES and non-PES patients. Statistical analysis showed that there were no significant differences in the infused total nucleated cells (TNC), CD34+ cells, CD3+ T cells, CD3-CD56+ NK cells, CD3-CD19+ B cells, or CD14+ monocyte dose when compared between the two groups (Fig. 2a). This result was in accordance with previous studies. Next, we tested whether there are differences in peripheral blood

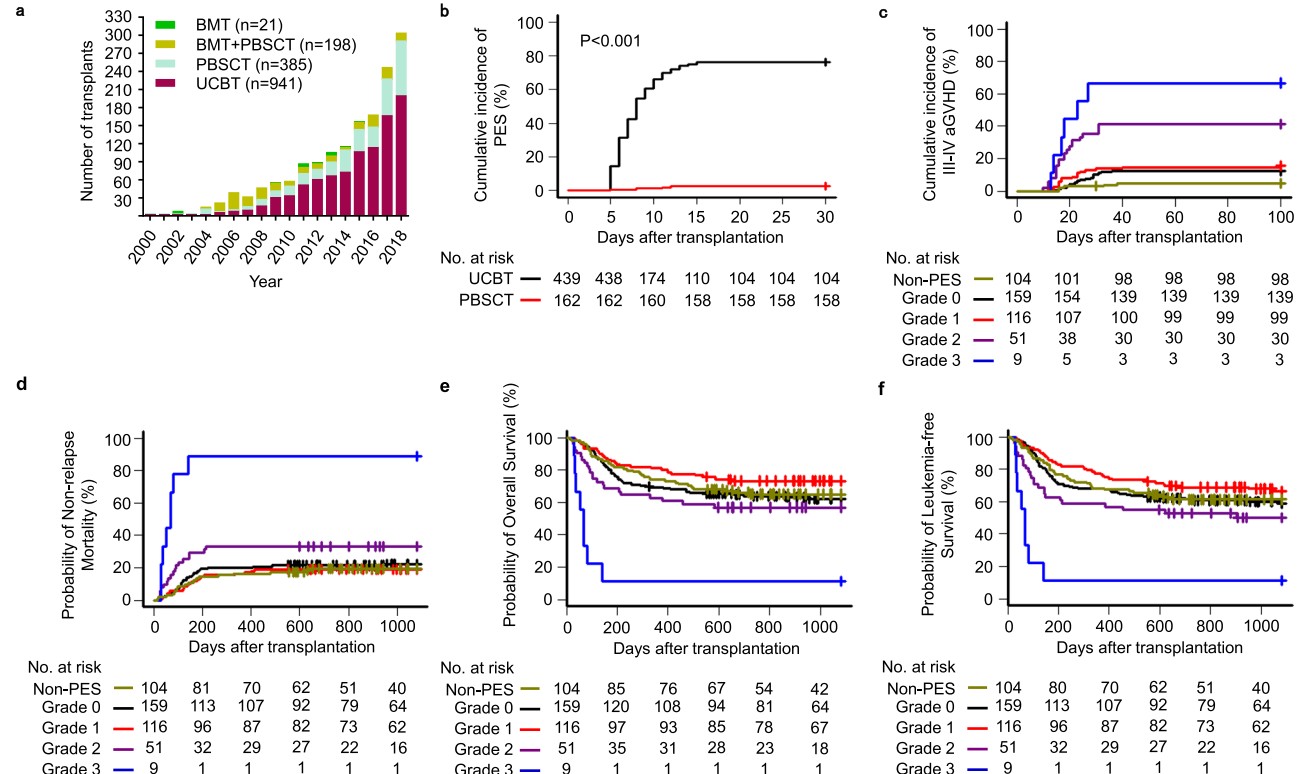

**Fig. 1 Outcomes after UCBT or PBSCT in patients with hematopoietic malignancies. a** Number of HSCT procedures performed per year. **b** The cumulative incidence of pre-engraftment syndrome in patients after UCBT or PBSCT. **c–f** The cumulative incidence of grades III–IV acute graft-versus-host disease at day 100, 3-year nonrelapse mortality, 3-year overall survival, and 3-year disease-free survival of patients with different PES grades: III–IV aGVHD (**c**), nonrelapse mortality (**d**), overall survival (**e**), and leukemia-free survival (**f**). The probabilities of PES, III–IV aGVHD and nonrelapse mortality as summarized by the cumulative incidence of competing events. The probabilities of overall survival and disease-free survival were calculated with Kaplan–Meier curves. Source data are provided as a Source Data file.

cells during the early stages after UCBT. The frequency of monocytes, but not T cells and NK cells, was significantly higher in patients with PES than non-PES, whereas B cells were barely detectable in peripheral blood (Fig. 2b–d, Supplementary Fig. 2a). Similar results were observed in the absolute number of peripheral blood cells; PES patients expanded with different kinetics. Furthermore, following the infusion of CB, PES patients had higher monocyte counts in the peripheral blood than non-PES patients, although the number of lymphocytes and neutrophils was comparable between the two groups (Fig. 2e–g). To address the origin of these findings, we investigated hematopoietic chimerism in the donors via peripheral blood analysis and polymerase chain reaction (PCR) amplification of short tandem repeats (STR–PCR) on days 7 and 14 after UCBT. As indicated, the severity of PES was highly correlated with donor chimerism on day 7 (Fig. 2h); however, there was no significant difference in the level of chimerism in PES patients by day 14 (Supplementary Fig. 3). Therefore, early chimerism was a significant risk factor for PES. Given the expansion of monocytes in the early stages after UCBT, we speculated that monocytes may play an important role in this inflammatory response. In the next stage of our work, we therefore characterized the functionality of monocytes.

**Monocytes derived from cord blood have inflammatory phenotypes.** In order to obtain mechanistic insights into clinical entity, we isolated CD14$^+$ monocytes from cord blood or peripheral blood for whole-genome RNA transcriptomic profiling analyses using microarray technology. Monocyte populations

were isolated using positive magnetic selection-based cell sorting. To evaluate whether human cord blood-derived monocytes, and peripheral blood-derived monocytes, represent functionally distinct subsets, we carried out further analysis on our microarray data. Both principal component analysis (Fig. 3a) and heat maps (Fig. 3b) demonstrated that monocytes derived from cord blood and peripheral blood stem cells exhibited distinct gene expression profiles. We specifically analyzed genes associated with the inflammatory process; remarkably, our data revealed that monocytes derived from cord blood showed upregulated levels of inflammatory cytokine transcripts, including GM-CSF, IL-6, and TNF-α (Fig. 3c). Volcano plot analysis revealed 6842 genes that were differentially expressed when compared with monocytes derived from cord blood and peripheral blood stem cells when considering a two-fold change threshold and $P < 0.05$. Notably, the gene expression levels of GM-CSF and IL-6 showed the most significant increase (Fig. 3d). Gene set enrichment analysis revealed that human cord blood-derived monocytes showed upregulation in terms of the activation of immune-response genes and the positive regulation of peptide-secretion genes (Fig. 3e–f). Collectively, these data showed that monocytes derived from cord blood represent an inflammatory population.

To test this hypothesis, we purified CD14$^+$ monocytes from human cord blood or peripheral blood stem cells by positive magnetic selection-based cell sorting. Using RT-PCR, we then measured mRNA expression of the pro-inflammatory mediators GM-CSF, IL-6, and TNF-α. Consistent with microarray analysis, cord blood monocytes expressed significantly higher levels of GM-CSF, IL-6, and TNF-α mRNA compared with monocytes

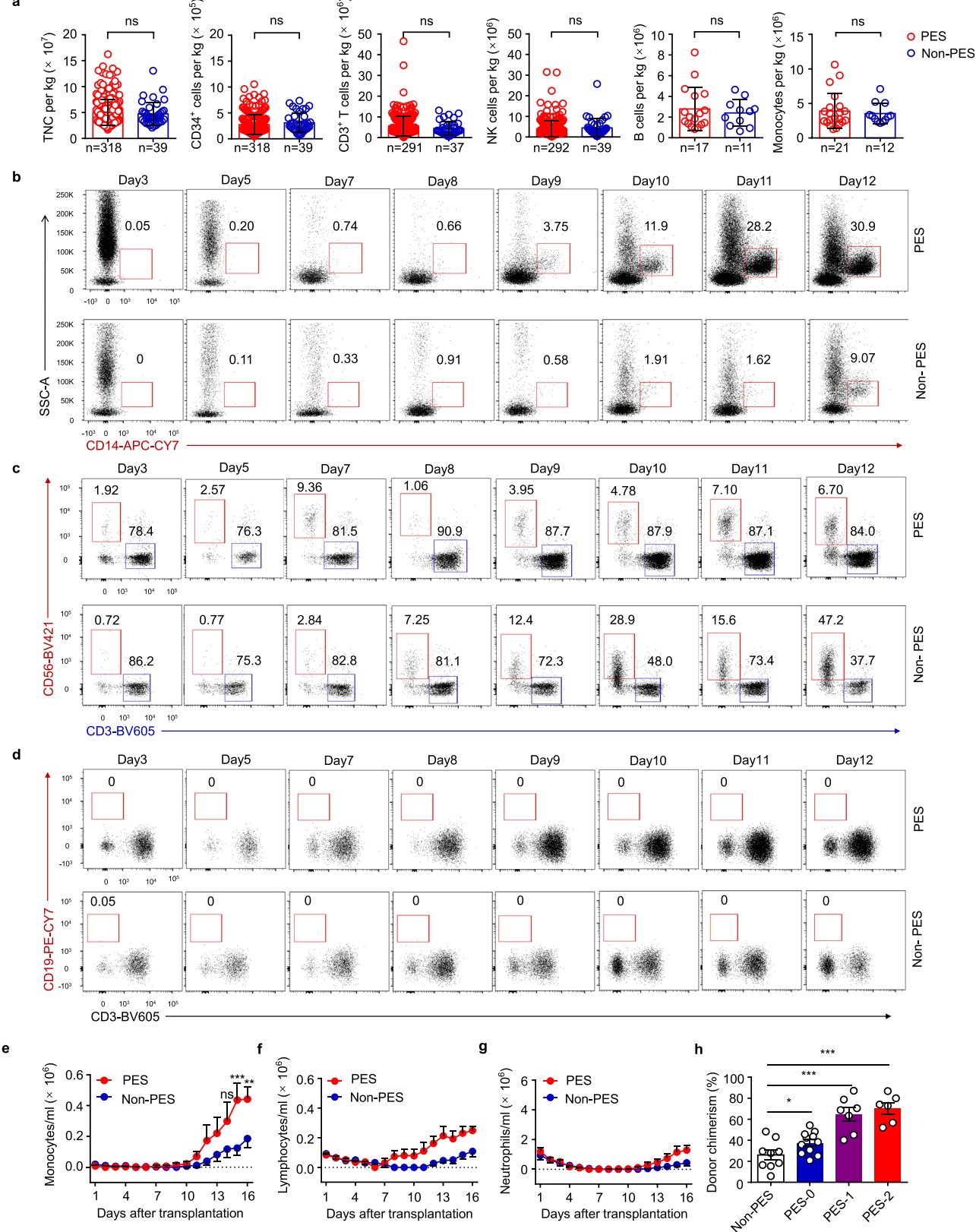

derived from peripheral blood stem cells (Fig. 3g). Intracellular staining results further highlighted a significant increase in IL-6 and TNF-α in monocytes derived from human cord blood, but not in T cells, NK cells, or B cells (Fig. 3h, i, Supplementary Fig. 2b, Supplementary Fig. 4a, b). Monocytes, T cells, NK cells and B cells did not express IFN-γ, IL-1β, and IL-17A in either

cord blood or peripheral blood (Supplementary Fig. 5a). Collectively, these results support the hypothesis that monocytes derived from cord blood secrete inflammatory cytokines. PES could therefore be attributed to a unique cytokine profile induced by the massive proliferation of cord blood-derived monocytes in the early posttransplantation period.

**Fig. 2 Expansion of cord blood cells during the early stages after UCBT. a** Number of cells infused into UCBT patients (PES, $n = 318, 318, 291, 292, 17$ and 21, respectively; Non-PES, $n = 39, 39, 37, 39, 11$, and 12, respectively). Mann–Whitney test (two-sided). **b** Frequency of monocytes in peripheral blood in the early stages after UCBT (PES, $n = 7$; Non-PES, $n = 6$). **c** Frequencies of T cells and NK cells in peripheral blood in the early stages after UCBT (PES, $n = 7$; Non-PES, $n = 6$). **d** Frequency of B cells in peripheral blood soon after UCBT (PES, $n = 7$; Non-PES, $n = 6$). **e** Dynamic changes of monocytes during the early posttransplant period after UCBT (PES, $n = 7$; Non-PES, $n = 6$). Holm–Sidak's multiple-comparison test without adjustment, $p = 0.1212, 0.0002$, and 0.0041, respectively. **f** Dynamic changes of lymphocytes during the early posttransplant period after UCBT (PES, $n = 7$; Non-PES, $n = 6$). **g** Dynamic changes of neutrophils during the early posttransplant period after UCBT (PES, $n = 7$; Non-PES, $n = 6$). **h** Donor chimerism in the peripheral blood of recipients on day 7 post UCBT ($n = 9, 11, 7$, and 6, respectively). One-way ANOVA (Non-PES vs. PES-0: mean difference = $-11.46$, 95% CI: $-23.80-0.885$, $p = 0.0676$; Non-PES vs. PES-1: mean difference = $-38.79$, 95% CI: $-52.63$ to $-24.95$, $p = 3e-6$; Non-PES vs. PES-2: mean difference = $-44.34$, 95% CI: $-58.81$ to $-29.87$, p = 7.71e-7). Each symbol in (**a**) and (**h**) represents an independent individual. Data are shown as means ± standard error of the mean (SEM). TNC total nuclear cells; n.s. not significant, *$p < 0.05$, **$p < 0.01$, and ***$p < 0.001$. Source data are provided as a Source Data file.

**GM-CSF drives the inflammatory signature of cord blood-derived monocytes**. Previous research has shown that monocytes are sensitive to GM-CSF, and that GM-CSF signaling controls a pathogenic expression signature in monocytes. We confirmed that monocytes, but not T cells and NK cells, were the main source of GM-CSF in cord blood; in contrast, GM-CSF+ monocytes were barely detectable in peripheral blood stem cells (Fig. 3j). GM-CSF+ monocytes were significantly increased in CB compared with PBSCs (Supplementary Fig. 5b). Similar results were obtained in cell culture supernatant because substantially higher amounts of GM-CSF were also found in culture supernatant, as measured by enzyme-linked immunosorbent assay (ELISA) (Supplementary Fig. 5c). Because monocytes express the GM-CSF receptor (Supplementary Fig. 6), and all patients receive G-CSF on day six after UCBT, we cultured freshly isolated monocytes from cord blood or peripheral blood stem cells by apheresis and stimulated them with GM-CSF, G-CSF, or lipopolysaccharide (LPS) for 6 h in vitro, respectively. We found that GM-CSF, but not G-CSF, could increase the expression of IL-6 in both groups, although the responsivity to LPS was similar between the two groups. Interestingly, we also found that even unstimulated CB-derived monocytes produced IL-6, possibly due to autocrine GM-CSF (Fig. 4a, Supplementary Fig. 2c). Measurement of IL-6 protein concentration in the cell culture supernatant further demonstrated that CB-derived monocytes produced more IL-6 than PBSC-derived monocytes (Fig. 4b, c). Collectively, these results indicate that GM-CSF promotes the expression of inflammatory cytokines in monocytes. It is possible that CB-derived monocytes contribute to the pathophysiology of PES via the production of GM-CSF and IL-6.

**Monocytes produced IL-6 in patients with PES**. Given that GM-CSF plays a vital role in the activation of monocytes, and because we observed a sizeable population of monocytes in PES patients during the early posttransplant period (Figs. 2b, 2e), we next tried to ascertain whether GM-CSF plays a role in the pathophysiology of PES. We therefore investigated dynamic changes of GM-CSF, IL-6, MCP-1 (also known as CCL2), and IL-8 (also known as CXCL8) levels in plasma. IL-6 levels were consistent with those of GM-CSF; levels of both of these proteins peaked around the time that PES patients first showed symptoms but then returned to baseline levels. In contrast, GM-CSF and IL-6 were almost always barely detectable in the plasma of non-PES patients, and we found that the levels of plasma IL-6 in patients on the day of PES were significantly higher than that of Non-PES patients (Fig. 4d, Supplementary Fig. 7a). Furthermore, the severity of PES is highly correlated with IL-6 levels (Supplementary Fig. 7b). Levels of MCP-1 and IL-8 were consistent between the two groups (Fig. 4d). To further confirm whether monocytes were the biological source of IL-6, we performed intracellular staining by flow cytometry. We observed that monocytes derived from PES patients contained large numbers of IL-6-producing cells in peripheral blood. In contrast, there are no IL-6-producing cells in T cells and NK cells (Fig. 4g, Supplementary Fig. 2d). Furthermore, the level of IL-6 was highly correlated with GM-CSF (Fig. 4f).

However, the levels of TNF-α, IL-1β and IFN-γ were barely detectable in both groups (Fig. 4e). Collectively, these results indicate that IL-6 is produced by monocytes in PES patients, but not by T cells and NK cells.

**Blocking IL-6 can ameliorate PES**. At present, steroids treatment is the first-line therapy for PES[17–19,24], symptoms improve promptly in patients with mild PES following the initiation of intravenous systemic corticosteroid therapy, while patients with severe PES had no response even when the dosage of methylprednisolone was up to 2 mg/kg/day[18]. Retrospective studies have shown a high mortality rate in patients with severe PES (methylprednisolone, 2 mg/kg/day). In mechanistic studies, we found that IL-6 may be the cause of severe PES. To verify whether targeting IL-6 signals may potentially save patients with severe PES, we applied for a clinical trial using tocilizumab to block the IL-6 receptor at www.chictr.org.cn (Reference: ChiCTR1800015472), and we used the historical 2 mg/kg/day methylprednisolone subgroup as the control group for ethical considerations. The baseline information of patients with severe PES for the single-arm study and the historical controls' study is shown in Supplementary Table 4. This trial enrolled 11 patients who suffered steroid-refractory severe PES after a single-unit UCBT as a first HSCT. There was no evidence of infection or adverse effects of medication in any of these patients. All patients were given a comprehensive workup, including urine cultures, and blood cultures through peripheral and central lines. All patients had failed to respond to an empirical antibiotic therapy. Eligible patients were treated with 4–8 mg/kg of the anti-IL-6 receptor monoclonal antibody, tocilizumab, with additional corticosteroids[30,31]. The end point used to evaluate efficacy was response, defined as an obvious improvement in fevers and rash.

Our trial showed that the responsivity of PES to tocilizumab, in terms of fever and skin rash, was prevented by blocking the IL-6 receptor with tocilizumab (Fig. 5a–d). Furthermore, all of the patients obtained neutrophil engraftment and successfully recovered from hematopoiesis (Fig. 5e). The nonrelapse mortality was 0% (95% [CI], 0.0–0.0) among patients who were treated with tocilizumab and 36% (95% CI, 22.9–49.2) among patients in the retrospective group (Reference: ChiCTR-ONC-16009013), thus, resulting in a substantial extension to overall survival in tocilizumab group (Fig. 5f). All of the patients enrolled in our trial recovered and were discharged from hospital. Collectively, these results show that tocilizumab can reverse PES.

## Discussion

UCBT offers us a chance of treating patients with hematological malignancy who do not have a matched-sibling donor[2,12]. Ever

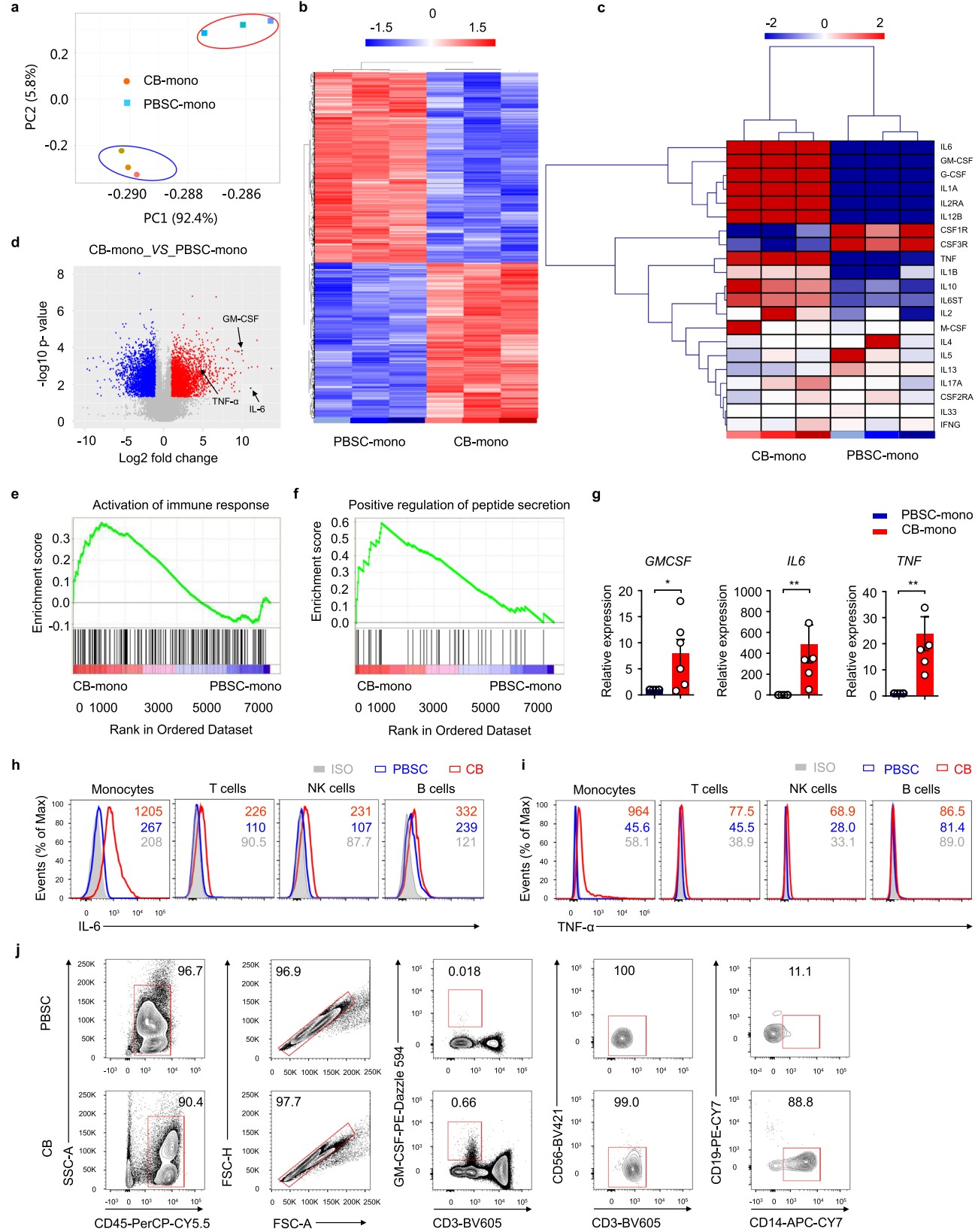

since the establishment of public cord blood banks, cord blood can be harvested from the umbilical cord of an anonymous newborn infant, and then cryopreserved. Stored cord blood is thus immediately available for transplantation purposes, and cord blood transplantation requires less-stringent HLA matching; this therefore makes it much easier to find suitable donors[1,32]. Another advantage of UCBT is the significantly lower rates of aGVHD and cGVHD, despite broader HLA disparity, but without compromising the GVL effect[33,34]. Furthermore, the probability of relapse is lower for UCBT than for PBSCT, or bone

**Fig. 3 Inflammatory characteristics of monocytes derived from umbilical cord blood. a** Principal component analysis of monocytes derived from CB (CB-mono) and PBSC (PBSC-mono). Red, CB-mono; blue, PBSC-mono. Each point represents a biologically independent sample. **b** Heat map representation of differentially-expressed genes (fold change >2, adjusted $P < 0.05$) between CB-mono and PBSC-mono. Each column represents an individual biological replicate. **c** Heat map representation of the expression of key proinflammatory and regulatory genes in CB-mono and PBSC-mono. Each column represents an individual biological replicate. **d** Comparison of microarray expression values in CB-mono and PBSC-mono. **e** GSEA plots showing that the activation of immune-response genes was upregulated in CB-mono. **f** GSEA plots showing that positive regulation of peptide-secretion genes was upregulated in CB-mono. **g** GM-CSF, IL-6, and TNF-α expression in CB-mono or PBSC-mono ($p = 0.0476$, 0.0022, and 0.0022, respectively), as assessed by RT-PCR. Data are representative of six experiments. Mann–Whitney test (two-sided). **h** Representative histograms of intracellular IL-6 in monocytes, T cells, NK cells, and B cells from peripheral blood stem cells or cord blood; $n = 16$ for peripheral blood stem cells; $n = 17$ for cord blood. **i** Representative histograms of intracellular TNF-α in monocytes, T cells, NK cells, and B cells from peripheral blood stem cells or cord blood; $n = 12$ for peripheral blood stem cells; $n = 15$ for cord blood. **j** Flow cytometry gating strategy utilized to identify GM-CSF$^+$ cells; PBSC, $n = 6$; CB, $n = 12$. The numbers in (**h**) and (**i**) show the mean fluorescence intensity (MFI) of IL-6 and TNF-α expression, respectively. Data in (**g**) are presented as means ± standard error of the mean (SEM). *$p < 0.05$, **$p < 0.01$. Source data are provided as a Source Data file.

marrow transplantation. Recent studies have reported discordant results, HLA–haploidentical stem cell transplantation with post-transplantation cyclophosphamide provides improved outcomes compared with ATG-containing UCBT[35–37]. Some studies have found a deleterious effect of exposure of ATG in UCBT, the results might be improved by omitting ATG from the conditioning regimen[38,39]. Further comparisons are warranted to better donor selection.

Although umbilical cord blood transplantation has many advantages, the incidence of PES remains high. A growing body of literature has reported PES and the effect of this syndrome on clinical outcomes. However, the mechanisms underlying the symptoms of PES remain poorly understood. At present, corticosteroids represent the first-line treatment for PES. Patients with severe PES should be treated with a large amount of corticosteroids because of the potential for progression, or the risk of life-threatening toxicity. However, corticosteroids may also cause a variety of complications, including osteoporosis and aseptic bone necrosis, and can even lead to graft failure[32]. Furthermore, some patients are steroid-refractory, and in the absence of other treatment options, have a high mortality rate. Here we report that inflammatory monocytes derived from cord blood play a critical role in the pathogenesis of PES (Supplementary Fig. 8). However, a recent study suggests that the main systemic sources of IL-6 are mostly in the tissue[40]. Interestingly, we found that after PBSCT, patients almost did not develop PES. This is a unique clinical manifestation that occurs after UCBT. In our study, we provide evidence that the activation of monocytes plays a critical role in this process in that these monocytes express IL-6. However, it is possible that tissue-derived IL-6 may also play a contributory role, at least in part. Although PES can occur as early as day 5 after UCBT, we can only detect large numbers of monocytes in peripheral blood one week after the occurrence of PES, possibly due to the local response of monocytes in the tissue prior to release into the peripheral blood. Importantly, these findings offer a new therapeutic strategy with which to tackle severe PES; in particular, we found that toxic effects can be mitigated by blockade of the IL-6 receptor. These clinical results suggest that tocilizumab may effectively improve signs and symptoms of PES. Coincidentally, we noted that a study sponsored by Memorial Sloan Kettering Cancer Center using tocilizumab to ameliorate aGVHD and early toxicity after double UCBT is ongoing (https://clinicaltrials.gov (NCT03434730)). We will pay close attention to the results of the study. Given that GM-CSF is upstream of IL-6, blockade of GM-CSF would be potentially more effective, which is what we are trying to do.

Interestingly, we also observed a population of GM-CSF$^+$ monocytes in human cord blood, but not in the peripheral blood. We demonstrated that monocytes derived from human cord blood exhibit inflammatory characteristics. It is possible that monocytes derived from cord blood are activated at the end of gestation just prior to labor[28,41,42]. The GM-CSF receptor complex is mainly expressed in myeloid cells;[43,44] it is possible that monocytes are targeted by this cytokine during PES, and that GM-CSF allows monocytes in the cord blood graft to produce pro-inflammatory mediators. We believe that this is why PES commonly occurs following UCBT. Hematopoietic stem cells (HSCs) are known to express receptors for inflammatory cytokines, and certain aspects of inflammatory cytokine signaling can stimulate HSCs to proliferate directly[45–47]. Thus, the goal of PES management is to prevent severe, life-threatening PES while retaining mild PES to maintain the greatest chance of engraftment by the HSCs and provide a unique GVL effect[26]. Future work is now needed to identify specific factors that could predict severe cases of PES.

Despite such insight, we are well aware that our study has some limitations. In the mechanistic studies, the blood samples taken for the cytokine and cellular detection were collected daily within two weeks after UCBT. Due to ethical limitations, it was difficult to carry out experiments with a larger sample size. The small number of samples included in the analysis is due to sample limitations rather than selection bias. Given a small number of patients, further validation of our findings in a prospective investigation of a larger series of PES patients is warranted. PES is distinct from aGVHD based on onset timing, preceding neutrophil recovery. However, PES occasionally progressed and merged with aGVHD, despite clinical treatments, because cytokine storm associated with PES might trigger the development of aGVHD. As utilizing tocilizumab could prevent and treat aGVHD[48–50], a similar mechanism might be involved in the development of PES and aGVHD.

In summary, UCBT shows impressive clinical potential, but the effective application of this therapeutic requires clinicians to promptly recognize PES and manage this condition effectively. Our current study provides a deeper understanding of the mechanisms involved in PES and proposes a new treatment strategy for cases with steroid-refractory PES. This syndrome should be taken into account when physicians encounter patients with fever and skin rash before engraftment that is not attributable to infection. Corticosteroids are recommended for the treatment of mild PES patients, while severe cases should be additionally treated with tocilizumab. We anticipate that this study will improve the efficacy and safety of UCBT.

## Methods

**Patients.** This study evaluated outcomes in patients who had high-risk or recurrent refractory hematological malignancies and underwent transplantation between April 2001 and June 2017. Patients were eligible if adequate outcome data were available and if they underwent a myeloablative conditioning regimen, and received a single unrelated cord blood unit, or allogeneic peripheral blood stem cells. A total of 601 patients met our criteria (the characteristics of patients and transplants are

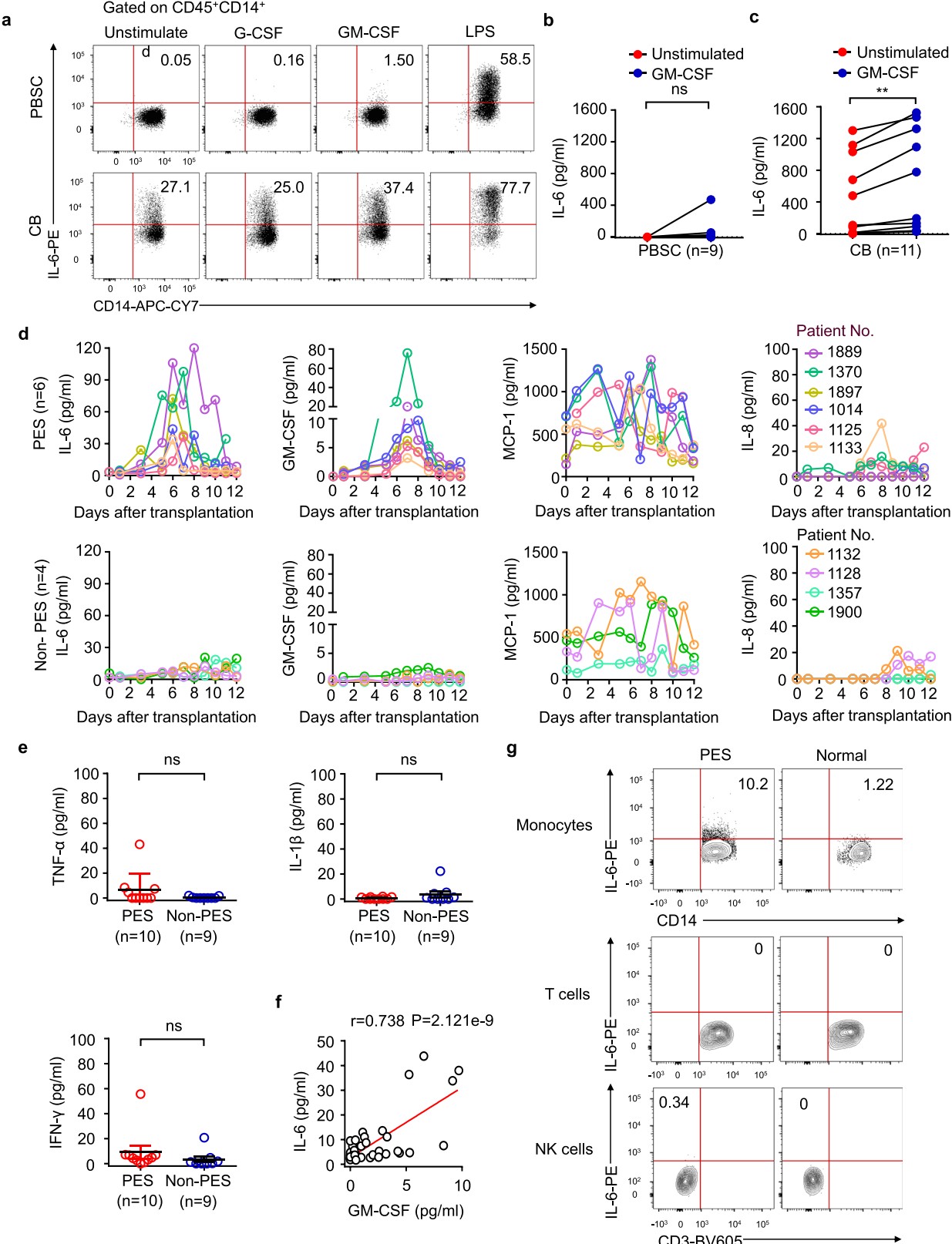

illustrated in Supplementary Table 2): 439 received cord blood (demographic and clinical characteristics of UCBT patients are shown in Supplementary Table 3) and 162 received peripheral blood stem cells. Intravenous G-CSF (filgrastim, 5–7 µg/kg/day) was commenced on day +6 and continued until the absolute neutrophil count exceeded $2.0 \times 10^9$/L for three consecutive days. The source of stem cells was determined mainly according to the consensus of allogeneic hematopoietic transplantation for hematological diseases in China, and in some cases according to patient preference.

**Subjects**. Study participants were recruited at the First Affiliated Hospital of the University of Science and Technology of China. Eligible patients with severe pre-engraftment syndrome, who are steroid-refractory and had a fever in excess of 38.3 °C for three consecutive days, had to be considered fit in order to receive tocilizumab (Tocilizumab Actemra, Roche). All patients underwent modified myeloablative conditioning chemotherapy or chemoradiotherapy. Patients or their guardians provided written informed consent. Nine patients received fludarabine (30 mg/m² for 4 days), busulfan (a total of 12.8 mg/kg, along with 0.8 mg/kg of

**Fig. 4 IL-6 production in patients with PES. a** Frequency of IL-6-producing CD14[+] monocytes within the CD45[+] population from peripheral blood stem cells or cord blood mononuclear cells during a 6-h culture period. Representative plots are shown from a total of nine independent experiments. **b**, **c** ELISA of IL-6 in supernatants from PBSC or CB during a 6-h culture period. PBSC, $n = 9$, CB, $n = 11$. Paired $t$ test (two-sided). **d** Dynamic changes in the plasma cytokine levels of patients with PES or without PES (PES, $n = 6$; Non-PES, $n = 4$). **e** Plasma cytokine levels in patients with or without PES. PES, $n = 10$; Non-PES, $n = 9$. Mann–Whitney test (two-sided). **f** Linear regression analysis of the association between the concentration of GM-CSF and IL-6 ($p = 2.121e-9$, $R = 0.738$, by using a two-sided Pearson correlation coefficient). **g** Frequencies of IL-6-producing monocytes, T cells, and NK cells within the CD45[+] populations from peripheral blood in patients with PES or healthy donors, $n = 3$. In order to compare means (PES vs. Non-PES), Mann-Whitney test (two-sided) was used in (**d**). Data are shown as means ± standard error of the mean (SEM). n.s., not significant, **$p < 0.01$. Source data are provided as a Source Data file.

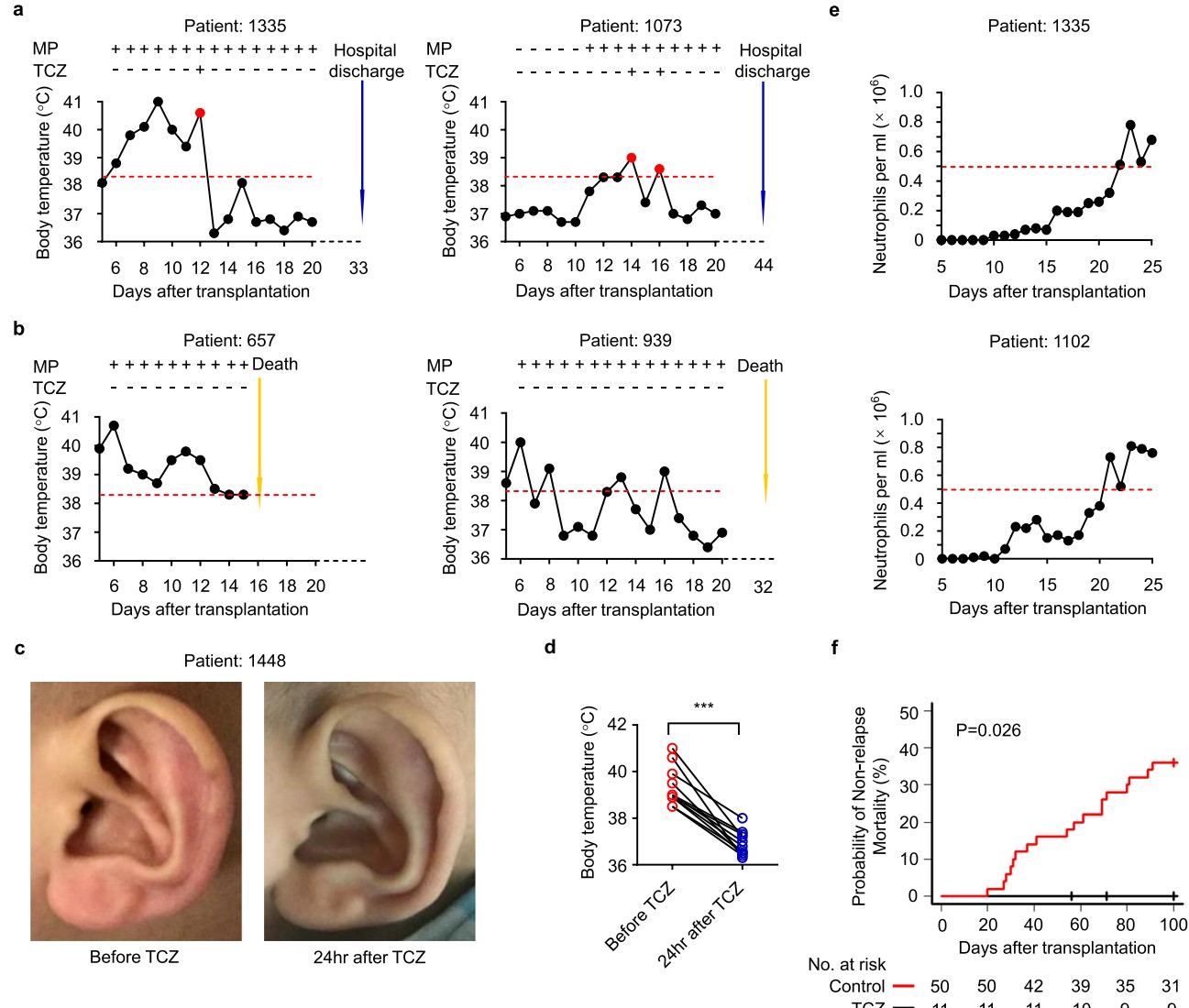

**Fig. 5 Clinical signs and symptoms before and after treatment with tocilizumab. a**, **b** Representative dynamic changes of body temperature in patients with PES. Treated with tocilizumab (**a**), treated without tocilizumab (**b**). **c** Representative images of skin rash from the ear of a PES patient before and after treatment with tocilizumab (similar results were obtained in all patients). **d** Body temperature of patients with PES before and after treatment with tocilizumab. Paired t test (two-sided) ($n = 11$, $p = 4e-6$). **e** Dynamic changes of neutrophils in PES patients treated with tocilizumab. **f** The probability of nonrelapse mortality in tocilizumab group and the retrospective group. The probability of nonrelapse mortality was performed by the cumulative incidence of competing events and Gray test. ***$p < 0.001$. TCZ, tocilizumab. Source data are provided as a Source Data file.

intravenous busulfan every 6 h for 4 days) plus 60 mg/kg of cyclophosphamide for 2 days. Two patients received total-body irradiation (a total of 12 cGy; 3 cGy twice a day for two days), cyclophosphamide (a total of 120 mg/kg administered as 60 mg/kg daily for 2 days) and cytarabine (2 g/m² twice a day for 2 days). The clinical trial was registered at www.chictr.org.cn (Reference: ChiCTR1800015472; Title: A single-arm, single-center clinical study of tocilizumab in the treatment of corticosteroid- unresponsive pre-engraftment syndrome patients after unrelated

cord blood transplantation). This study was approved by the Chinese Ethics Committee of Registering Clinical Trials (ChiECRCT-20180129).

**Human samples**. Mononuclear cells were isolated from umbilical cord blood (The First Affiliated Hospital of the University of Science and Technology of China) and prepared from buffy coats obtained from healthy infant donors by centrifugation

and a Ficoll system. Cord blood was layered on top of Ficoll (tbd science, # LTS1077) in a 50-mL tube and centrifuged at 500 x *g* for 30 min at room temperature. Buffy coats were collected, washed twice in phosphate-buffered saline (PBS), and finally resuspended. Peripheral blood stem cells were collected from the peripheral blood through a process known as apheresis. Donors were given daily subcutaneous injections of G-CSF, to help mobilize stem cells from the bone marrow and into the peripheral circulation.

**Cell culture and stimulation conditions**. Umbilical cord blood mononuclear cells and peripheral blood stem cells were cultured in RPMI Medium Modified (Hyclone, # SH30809.01) with 10% FBS (Gibco, # 1932562), with or without recombinant GM-CSF (100 ng/mL), G-CSF (100 ng/mL), or LPS (100 ng/mL) at 37 °C in a 5% $CO_2$ incubator for 6 h. Cells were then harvested for FACS or magnetic selection.

**Isolation of human monocytes**. Primary human cord blood monocytes were purified from umbilical cord blood mononuclear cells, while primary human peripheral monocytes were purified from peripheral blood stem cells. Cells were resuspended in buffer, a solution containing phosphate-buffered saline (PBS), pH 7.2, 0.5% bovine serum albumin, and 2 mM EDTA (SCRC, China, # 10009617). Then, $CD14^+$ cells were isolated by positive magnetic selection (CD14 MicroBeads, human, Miltenyi, # 130-050-201), in accordance with the manufacturer's instructions.

**Plasma sample collection**. Blood specimens were collected by venipuncture into 5-mL tubes containing heparin. Then, whole blood was centrifuged at 300 x *g* for 10 min at 4 °C. The top plasma layer was then aliquoted into tubes and immediately stored at −80 °C until analysis.

**ELISA**. Plasma and cell culture supernatant cytokines were measured using ELISA kits for human IL-6, human GM-CSF, human TNF-α, human IFN-γ, human IL-1β, human MCP-1, and human IL-8, in accordance with the manufacturer's instructions. Data were collected by measuring the absorbance at 450 nm and 630 nm with a plate reader (BioTek, ELX800).

**Flow cytometry**. Whole-blood samples were incubated directly with fluorescent antibodies for 30 min at 4 °C in the dark. Red blood cells (RBCs) were subsequently lysed using BD FACS[TM] lysing solution. Once lysed, the cells were washed twice in PBS, resuspended in 200 μl of PBS, and analyzed with a BD LSRII or LSRFortessa flow cytometer (BD Biosciences). For intracellular staining, mononuclear cells or peripheral blood stem cells were incubated with monensin (10 ng/mL, Sigma), with or without recombinant GM-CSF (100 ng/mL) or LPS (100 ng/mL) for 6 h at 37 °C in a 5% $CO_2$ incubator. Cells were then labeled for extracellular markers, fixed, permeabilized with [an eBioscienceTM] Foxp3/Transcription Factor Staining Buffer Set and stained for intracellular molecules. Human-specific fluorophore-conjugated antibodies for flow-cytometry staining are shown in Supplementary Table 5. These antibodies were all purchased from commercial vendors. Data were analyzed with FlowJo 10 software.

**RNA extraction and quantitative PCR**. To compare the expression of inflammatory factors in monocytes derived from umbilical cord blood and peripheral blood stem cells, monocytes were sorted directly into 1 ml of TRIzol reagent. Total RNA was then extracted in accordance with the manufacturer's guidelines. cDNA was synthesized using M-MLV Reverse Transcriptase and oligo dT primers (Sangon Biotech). cDNA was then analyzed for the relative expression of various inflammatory factors; this was carried out by real-time quantitative RT-PCR with TB Green Premix Ex Taq II, primers for target genes are shown in Supplementary Table 6. The levels of each gene were normalized to β-actin. Key reagents used in this study are shown in Supplementary Table 7.

**Microarray**. Whole-genome transcriptome analysis was carried out on monocytes (in triplicate) using an Agilent Human mRNA microarray (4 × 44 K). Microarray data were analyzed using Feature Extraction v10.7.1.1 (Agilent Technologies) and GeneSpring GX (Agilent Technologies). Microarray data were then uploaded to GEO (accession number: GSE128562).

**Statistical analysis**. Statistical analysis was performed using R software package (version 3.5.0), Prism six (GraphPad), and SPSS 25.0 software, and involved two-tailed unpaired t tests, paired t tests, the cumulative incidence of competing events, and Kaplan–Meier survival curves. Cumulative incidence curves were used in a competing-risk setting, with death treated as a competing event, to calculate the probability of PES, aGVHD, and cGVHD. Probabilities of disease-free survival and overall survival were estimated using the Kaplan–Meier method. The statistical test used for each figure is described in the corresponding figure legend. For multiple testing, we refer to the significance level α of 0.01.

**Reporting summary**. Further information on research design is available in the Nature Research Reporting Summary linked to this article.

## Data availability

Microarray data have been deposited in the National Center for Biotechnology Information Gene Expression Omnibus repository under accession number: GSE128562. The clinical characteristics of all patients included in the present study are shown in Supplementary Table 2–4. Antibodies used were shown in Supplementary Table 5. The data that support the findings of this study are available from the corresponding author upon request. Source data are provided with this paper.

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

## Acknowledgements

This study was supported by the key project of the National Natural Science Foundation of China (81788101), Chinese National Natural Science Foundation (81470350) and the Fundamental Research Funds for the Central Universities (WK9110000001) in China.

## Author contributions

L.J. designed and performed experiments, analyzed, and interpreted the data. H.L. and X. Zhu conducted the clinical trial. Y.Z., B.F. and X.Zheng provided advice. K.S., B.T., Y.W., and J.Z provided study materials or patients. R.S. established the techniques of FACS and interpreted the data. Z.T. provided strategic planning and interpreted some data. Z.S. and H.W. designed the study, supervised the research, interpreted the data, and revised the paper.

## Competing interests

The authors declare no competing interests.
