## [Peer Review File · Nature Communications]

Reviewers' comments:

Reviewer #1 (Remarks to the Author):

This is a well written manuscript about pre-engraftment syndrome with well thought out experiments. My comments are below

1. I am concerned about the overall reported incidence of pre-engraftment syndrome which seems unusually high. Additionally I am unsure why the authors chose to grade the pre-engraftment syndrome. It is unclear to me if the plasma values of IL6, TNF α , IL8 levels, IFNG levels are being compared between all cases of pre-engraftment syndrome versus no engraftment syndrome or just Grade 3 pre-engraftment syndrome versus no engraftment syndrome
2. Is there a difference in cytokine profiles between the grades of the pre-engraftment syndromes? If yes, what is the explanation for this difference?
3. The authors mention that T cells and NK cells do not express IL6 or GM-CSF compared to monocytes in PES, but there would likely be no T-cells or NK cells detected so early after stem cell infusion. What are the cell event numbers used in flow cytometry for the intracellular IL6 expression on T-cells and NK cells that have been used to make these comparisons.
4. It is hard to make the connection between the IL6 levels in cord blood monocyte and the peripheral blood monocytes expressing elevated IL6 levels in patients- are the authors suggesting that the monocytes infused in the cord blood have persisted in the donor leading to elevated IL6 levels and PES or are these new donor stem cell derived monocytes which also seem to express elevated IL6 levels?
5. Have the authors performed in vitro experiments with G-CSF in addition to GM-CSF? Clinically, patients receive G-CSF and perhaps that could alter the differences observed in the cytokines measured from cord blood stem cells and peripheral blood stem cells?
6. Could the authors explain what they mean by engraftment syndrome which is refractory to steroids? What doses and frequency of steroids are allowed before PES is considered refractory?

Reviewer #2 (Remarks to the Author):

In this study Jin et al describe the inflammatory mediators during pre-engraftment syndrome during UCBT. They define PES as being characterized by early engraftment/chimerism and monocyte expansion in blood. CD14⁺ monocytes from CB are shown to be hyperinflammatory (cf adult PBSC) with exaggerated GM-CSF and IL-6 secretion. CB monocytes expressed higher levels of the GM-CSFR than adult PBSC and responded to GM-CSF with higher levels of IL-6 secretion. GM-CSF and IL-6 were increased in sera in selected patients and intervention with TCZ appeared effective in patients with refractory PES.

The manuscript is well written and the finding is potentially novel and clinically important.

Major issues:

- 1) It is not clear how data has been selected. There are a large number of patients yet n values are small in the mechanistic studies and are not defined in most Figures. Please define all n values, how patients were selected if only a subset are analyzed and how is missing data imputed or dealt with?
- 2) Rather than just showing representative histograms please also quantify data over a larger patient cohort to allow the reader to understand reproducibility. Fig 3G, 3H, 4G are some examples.
- 3) The IL-6 levels in Fig 4D are not particularly high and only 4 patients are shown. Please show a larger cohort and show all data available, not selected patients.
- 4) How do the authors distinguish PES from acute GVHD? The severity index is measuring features of acute GVHD, are their biopsies to exclude GVHD, there must be an overlap with GVHD as suggested in SFig 1 and this should be discussed, as could the clinical data using TCZ to prevent and

treat acute GVHD that is published. Presumably blockade of GM-CSF would also be effective and potentially more effective since it is upstream of IL-6; clinical reagents are available to block GM-CSF and this could also be discussed.

5) The authors make definitive statements about sources of IL-6 and GM-CSF after UCBT based on analysis of blood only. The main systemic sources of the cytokine are almost certainly in tissue, and perhaps are not even donor (e.g. Wilkinson AN et al, Blood 2019). Thus conclusions based on blood could be misleading and the authors interpretations should reflect these limitations.

6) A diagram outlining the author's putative pathway of cytokine dysregulation in PES would be helpful.

7) A statistical review would be helpful, t-tests are undertaken on very small data sets where data is unlikely to be normally distributed. Non-parametric tests would be more appropriate if this is the case.

Reviewers' comments:

Reviewer #1 (Remarks to the Author):

This is a well written manuscript about pre-engraftment syndrome with well thought out experiments. My comments are below

Author response: Thank you for your comments and suggestions.

1. I am concerned about the overall reported incidence of pre-engraftment syndrome which seems unusually high. Additionally, I am unsure why the authors chose to grade the pre-engraftment syndrome. It is unclear to me if the plasma values of IL6, TNFa, IL8 levels, IFNG levels are being compared between all cases of pre-engraftment syndrome versus no engraftment syndrome or just Grade 3 pre-engraftment syndrome versus no engraftment syndrome.

Author response: We apologize for causing confusion. The incidence of pre-engraftment syndrome (PES) in our study was 76.3%. In order to compare our data with that obtained from other centers, we reviewed a range of studies (as shown in Table 1 below). We found that the incidence of PES after unrelated cord blood transplantation (UCBT) has previously been reported to be high, ranging from 20% to 86.8%.

Table 1. The incidence of PES in different transplant centers

References	Number of patients	% of patients developing PES
Kishi, et al ^{18*} .	45	78
Lee, et al ^{19*} .	14	23.1
Brownback, et al ^{20*} .	44	50
Park, et al ^{21*} .	381	26.8
Narimatsu, et al ^{22*} .	77	36
Frangoul, et al ^{23*} .	326	20
Patel, et al ^{24*} .	52	31

Wang, et al ^{25*} .	81	63
Kanda, et al ^{26*} .	57	77
Isobe, et al ^{27*} .	138	86.8

Abbreviations: PES, pre-engraftment syndrome. * The numbers given after each of the cited references refer to those in the reference list of our manuscript.

The severity of pre-engraftment syndrome was graded according to the clinical symptoms of the patients in order to conduct stratified intervention. We chose to grade pre-engraftment syndrome to maximize the chances of achieving therapeutic benefit while minimizing the risk of life threatening complications related to pre-engraftment syndrome. The goal of PES management is to prevent severe, life-threatening PES, while retaining mild PES so as to maintain the greatest chance of engraftment by the hematopoietic stem cells.

We compared the plasma values of IL-6, TNF- α , IL-8, and IFN- γ between all cases of PES *versus* those without PES (**Supplementary fig. 7a, Fig. 4e**).

Supplementary fig. 7a Plasma IL-6 levels were significantly higher in patients with PES than those without PES. Data are presented as mean \pm SEM. Mann-Whitney test. ****p < 0.0001.

Fig. 4e Plasma cytokine levels in patients with or without PES. Data are shown as

means \pm SEM. n.s., not significant.

2. Is there a difference in cytokine profiles between the grades of the pre-engraftment syndromes?

Author response: Thank you. We appreciate this important suggestion. We measured IL-6 levels in plasma between different grades of PES and performed statistical analysis. We found that the severity of PES was highly correlated with IL-6 levels (**Supplementary fig. 7b**).

Supplementary fig. 7b Plasma IL-6 levels were significantly higher in patients with severe PES.

If yes, what is the explanation for this difference?

Author response: It appears that the cord blood donors can be divided into no/low responders and responders that already show elevated levels in unstimulated conditions. We can therefore hypothesize that the individual functional properties of monocytes in specific cord blood grafts may predispose the receiving patient to PES. (**Fig. 4c**).

Fig. 4c. Heterogeneous expression of IL-6 in cord blood donors. ELISA of IL-6 in supernatants from CB during a 6h culture period (n=11).

3. The authors mention that T cells and NK cells do not express IL6 or GMCSF compared to monocytes in PES, but there would likely be no T-cells or NK cells

detected so early after stem cell infusion. What are the cell events numbers used in flow cytometry for the intracellular IL6 expression on T-cells and NK cells that have been used to make these comparisons.

Author response: Thank you. We appreciate this comment. With regards to intracellular IL-6 expression analysis, we acquired 30 000 events for leukocytes. However, for T cells and NK cells, we only acquired 1500 events and 700 events, respectively. In the early stages after unrelated cord blood transplantation, the number of T cells and NK cells was very small. Furthermore, the expression of IL-6 in these cells was very low or undetectable. Our study found that IL-6 was mainly produced by monocytes.

4. It is hard to make the connection between the IL6 levels in cord blood monocyte and the peripheral blood monocytes expressing elevated IL6 levels in patients- are the authors suggesting that the monocytes infused in the cord blood have persisted in the donor leading to elevated IL6 levels and PES or are these new donor stem cell derived monocytes which also seem to express elevated IL6 levels?

Author response: Thank you. We appreciate this comment. Patients usually develop PES as early as 5 days post-transplantation. Recent studies show that it takes two weeks for hematopoietic stem cells to differentiate into monocytes, and expand further at 3 weeks and beyond (Upadhaya et al., J. Exp. Med., 2018). However, the median time of engraftment was 21 days (Isobe et al., Biol Blood Marrow Transplant, 2019; Konuma et al., Biol Blood Marrow Transplant, 2017). At such an early stage after UCBT, it seems impossible that monocytes in the peripheral blood are derived from stem cells. We suggest that monocytes derived from cord blood responded to GM-CSF with high levels of IL-6 secretion and undergo rapid expansion in the recipient, thus leading to elevated levels of IL-6 levels in the peripheral blood (**Fig. 4c**).

5. Have the authors performed in vitro experiments with GCSF in addition to GMCSF? Clinically, patients receive GCSF and perhaps that could alter the differences observed

in the cytokines measured from cord blood stem cells and peripheral blood stem cells?

Author response: We apologize for causing confusion. Yes, we have performed *in vitro* experiments with G-CSF (**Fig. 4a**). We found that G-CSF did not increase the expression of IL-6 in monocytes, and that this was the case in both groups.

Fig. 4a GM-CSF significantly increased the expression levels of IL-6 in monocytes. Data represent the frequency of IL-6-producing CD14⁺ monocytes within the CD45⁺ population from peripheral blood stem cells or cord blood mononuclear cells during a 6 h culture period. Representative plots from a total of nine independent experiments are shown.

6. Could the authors explain what they mean by engraftment syndrome which is refractory to steroids? What doses and frequency of steroids are allowed before PES is considered refractory?

Author response: Thank you for your insightful suggestion. For steroid-refractory PES patients, the dose of methylprednisolone is usually 2 mg/kg/day; pediatric patients receive up to a maximum of 3mg/kg/day. If the manifestations of PES in any organ worsen over 3 days of treatment, or if the symptoms do not improve after 5 days on 2 mg/kg/day of methylprednisolone, we would consider a patient as being steroid-refractory and would use tocilizumab therapy. The doses of steroids used in our study refer to standard first-line systemic therapy for acute GVHD, as recommended by the

American Society of Blood and Marrow Transplantation (Martin et al., Biol Blood Marrow Transplant, 2012).

Reviewer #2 (Remarks to the Author):

In this study Jin et al describe the inflammatory mediators during pre-engraftment syndrome during UCBT. They define PES as being characterized by early engraftment/chimerism and monocyte expansion in blood. CD14⁺ monocytes from CB are shown to be hyperinflammatory (cf adult PBSC) with exaggerated GM-CSF and IL-6 secretion. CB monocytes expressed higher levels of the GM-CSFR than adult PBSC and responded to GM-CSF with higher levels of IL-6 secretion. GM-CSF and IL-6 were increased in sera in selected patients and intervention with TCZ appeared effective in patients with refractory PES.

The manuscript is well written and the finding is potentially novel and clinically important.

Author response: Thank you. We appreciate your comments and suggestions.

Major issues:

1) It is not clear how data has been selected. There are a large number of patients yet n values are small in the mechanistic studies and are not defined in most Figures. Please define all n values, how patients were selected if only a subset are analyzed and how is missing data imputed or dealt with?

Author response: We apologize for causing confusion. In the mechanistic studies, our human data and statistical analyses were not sufficiently clear. In the revised manuscript, the number of samples in each group is now shown in either the figures or figure legends. In terms of our analysis of clinical outcome, we evaluated outcomes in patients who had high-risk or recurrent refractory hematological malignancies and underwent transplantation between April 2001 and June 2017. Patients were eligible if adequate outcome data were available, if they underwent a myeloablative conditioning regimen, and if they received a single unrelated cord blood unit or allogeneic peripheral blood stem cells (see **Methods, Page 14, Line 307-339**).

2) Rather than just showing representative histograms please also quantify data over a larger patient cohort to allow the reader to understand reproducibility. Fig 3G, 3H, 4G are some examples.

Author response: Thank you very much for this valuable suggestion. In the revised manuscript, we have analyzed the figures statistically and present analysis of data from each sample (e.g. **Supplementary fig. 4a, 4b**). For example, as shown in Fig. 3h, Supplementary fig. 4a, FACS shows representative histograms of intracellular IL-6 in monocytes, T cells, NK cells, and B cells from the peripheral blood stem cell or cord blood; n=16 for peripheral blood stem cells; n=17 for cord blood. Supplementary Fig. 4a shows the statistical analysis of the results displayed in Fig. 3h.

Fig. 3h Representative histograms of intracellular IL-6 in monocytes, T cells, NK cells, and B cells from peripheral blood stem cell or cord blood (PBSC, n=16; CB, n=17).

Supplementary fig. 4a Statistical data generated by the MFI of IL-6 from mononuclear cells from peripheral blood stem cells (PBSC) or cord blood (CB), n = 16 and 17, respectively. Data are presented as mean \pm SEM. Mann-Whitney test. ****p < 0.0001.

3) The IL-6 levels in Fig 4D are not particularly high and only 4 patients are shown. Please show a larger cohort and show all data available, not selected patients.

Author response: Thank you for this valuable suggestion. In order to study the dynamic changes in plasma IL-6 levels in patients after UCBT, we acquired blood samples on a daily basis. For ethical reasons, we required patient consent. None of the patients shown in Fig. 4d had a documented infection. Although China is suffering from a COVID-19 epidemic, we have still managed to add 3 additional cases (PES, n=6; Non-PES, n=4). Because of the need for epidemic prevention and control, we clearly cannot continue to collect more cases, and hope that you can understand our position and support our current inability to proceed (**Fig. 4d**). In addition, we measured IL-6 levels on the day of PES and compared these levels with those of non-PES patients. We found that plasma IL-6 levels were significantly higher in PES patients than in non-PES patients (**Supplementary fig. 7a**).

Fig. 4d Dynamic changes in the plasma cytokine levels of patients with PES or without PES.

Supplementary fig. 7a Plasma IL-6 levels were significantly higher in patients with PES than without PES. Data are presented as mean \pm SEM. Mann-Whitney test. **** p < 0.0001.

4) How do the authors distinguish PES from acute GVHD? The severity index is

measuring features of acute GVHD, are their biopsies to exclude GVHD, there must be an overlap with GVHD as suggested in SFig 1 and this should be discussed, as could the clinical data using TCZ to prevent and treat acute GVHD that is published. Presumably blockade of GM-CSF would also be effective and potentially more effective since it is upstream of IL-6; clinical reagents are available to block GM-CSF and this could also be discussed.

Author response: Thank you for your insightful suggestion. Patients generally develop PES as early as 5 days post-transplantation; the median day of engraftment was day 21 (Isobe et al., Biol Blood Marrow Transplant, 2019; Konuma et al., Biol Blood Marrow Transplant, 2017). The symptoms of PES typically appear 7 or more days before neutrophil engraftment after UCBT; in contrast, aGVHD occurs after engraftment. In accordance with your suggestion, we have added further comments to the relevant part of the discussion (**Page 13, Line 291-295**).

In the revised manuscript, we also consider the monoclonal antibody that targets GM-CSF. We appreciate this important point from the reviewer and have provided additional discussion relating to this point (**Page 12, Line 274-275**).

5) The authors make definitive statements about sources of IL-6 and GM-CSF after UCBT based on analysis of blood only. The main systemic sources of the cytokine are almost certainly in tissue, and perhaps are not even donor (e.g. Wilkinson AN et al, Blood 2019). Thus conclusions based on blood could be misleading and the authors interpretations should reflect these limitations.

Author response: We appreciate your interest in the sources of IL-6 and GM-CSF, and thank you for your kind suggestion. We have now cited this literature and provided further discussion (**Page 12, Line 262-268**). This is an interesting question. While a recent study suggests that the main systemic sources of IL-6 are mostly in the tissue, we found that after peripheral blood stem cell transplantation, patients did not develop PES. This is a unique clinical manifestation that occurs after UCBT. In our study, we provide evidence that the activation of monocytes plays a critical role in this process in

that these monocytes express IL-6. However, it is possible that tissue-derived IL-6 may also play a contributory role, at least in part.

6) A diagram outlining the author's putative pathway of cytokine dysregulation in PES would be helpful.

Author response: Thank you. We appreciate your important suggestion. We have now included a diagram in the revised manuscript (**Supplementary fig. 8**).

Supplementary fig. 8 Monocytes derived from cord blood possess inflammatory characteristics, and secrete both GM-CSF and IL-6. Cord blood monocytes expressed high levels of GM-CSFR and responded to GM-CSF with high levels of IL-6 secretion. Subsequently, these cells undergo rapid expansion in the recipient. Levels of both GM-CSF and IL-6 were increased in the sera of PES patients. Intervention with tocilizumab (TCZ), the monoclonal antibody that targets the IL-6 receptor, is an effective treatment for patients with refractory PES.

7) A statistical review would be helpful, t-tests are undertaken on very small data sets where data is unlikely to be normally distributed. Non-parametric tests would be more appropriate if this is the case.

Author response: Thank you. We appreciate your constructive comments. In the revised manuscript, we have revised the statistical methods in accordance with your suggestion (e.g. **Fig. 2a**, **sFig. 4**, **sFig. 5b**, **5c**, **Fig. 4e**).

Reviewers' comments:

Reviewer #1 (Remarks to the Author):

My questions have been adequately addressed by the authors

Reviewer #2 (Remarks to the Author):

Comments have been adequately answered. I assume the smaller n values of cytokine and cellular data represent sample limitations rather than selection bias but this was not explicitly stated.

Reviewer #3 (Remarks to the Author):

This manuscript reports results from two studies. The first is a retrospective cohort study assessing the effect of monocytes from peripheral blood stem cells and cord blood in 601 patients with and without pre-engraftment syndrome (PES). The second study is a single-arm clinical study assessing outcome of 11 participants who received tocilizumab.

I am not an expert in this clinical area therefore I am unable to comment on the originality of the study.

My comments of the manuscript are primarily on the statistical aspect of the research.

Major comments:

- My main concern of the results report is the lack of consideration of the potential confounding issues that could arise due to the design of the study, which could lead to bias in the results reported in the manuscript. For example, without any baseline information between the cord blood and peripheral blood stem cells transplantation (PBSCT) recipients, it is difficult to interpret if the difference in the incidence of PES was due to any underlying difference at baseline. The number of patients who had PBSCT were far fewer than those who underwent UCBT, therefore it wasn't clear if PBSCT could only be eligible in a selective group of patients.
- Similarly, there were no baseline information about patients with and without PES and PES patients and whether adjusting for confounding is required.
- Figure 2a to Figure 2h – number included in the analysis were small and would like to know if it was because due to data availability or were randomly selected. A lot of analyses carried out were performed (including multiple outcomes and multiple time point analyses) and would be good to indicate if type I error were accounted for in these comparisons, and justify if not.
- I am probably not too worried about using t-test for small sample data but I think some of the data reported in this manuscript were skewed.
- If comparisons between groups were made, then if possible, the difference between the group and corresponding 95% CI should be reported rather than by group results.
- The single arm study was reported briefly in the manuscript. There was no justification on sample size and the rationale of why this study did not have a control group but was compared with historical controls from another observational study, which no information was provided about this study. Again, no baseline information were reported and therefore difficult to assess the comparability between the two groups. There were 3 PES subgroups who received different level of intervention in the observational study and it isn't clear which PES subgroup was used as the control group and why. I suggest the authors should follow the CONSORT statement as far as possible if the authors felt this was a clinical trial.
- In the Discussion section, the authors have stated "We successfully showed that tocilizumab can effectively improve signs and symptoms of PES". I find this statement is too strong and the treatment recommendation is worryingly flawed. based on findings from a very small single arm study and compared with historical "controls".

Reviewers' comments:

Reviewer #1 (Remarks to the Author):

My questions have been adequately addressed by the authors.

Author response: Thank you very much for your support and encouragement. Your insightful suggestions are very helpful to our article.

Reviewer #2 (Remarks to the Author):

Comments have been adequately answered. I assume the smaller n values of cytokine and cellular data represent sample limitations rather than selection bias but this was not explicitly stated.

Author response: We appreciate your understanding and support. There were hundreds of samples in our retrospective study. The blood samples taken for the cytokine and cellular detection were collected daily within two weeks after UCBT. Due to ethical limitations, it was difficult to carry out experiments with a larger sample size. We appreciate your important suggestion, and we have provided additional discussion relating to this point (**Page 13, Line 294-298**).

Reviewer #3 (Remarks to the Author):

This manuscript reports results from two studies. The first is a retrospective cohort study assessing the effect of monocytes from peripheral blood stem cells and cord blood in 601 patients with and without pre-engraftment syndrome (PES). The second study is a single-arm clinical study assessing outcome of 11 participants who received tocilizumab.

I am not an expert in this clinical area therefore I am unable to comment on the originality of the study.

My comments of the manuscript are primarily on the statistical aspect of the research.

Author response: Thank you. We appreciate your comments and suggestions.

Major comments:

1. My main concern of the results report is the lack of consideration of the potential confounding issues that could arise due to the design of the study, which could lead to bias in the results reported in the manuscript. For example, without any baseline information between the cord blood and peripheral blood stem cells transplantation (PBSCT) recipients, it is difficult to interpret if the difference in the incidence of PES was due to any underlying difference at baseline. The number of patients who had PBCST were far fewer than those who underwent UCBT, therefore it wasn't clear if PBSCT could only be eligible in a selective group of patients.

Author response: We apologize for causing confusion. The baseline information between the unrelated cord blood transplantation (UCBT) and peripheral blood stem cell transplantation (PBSCT) recipients is now shown below (Supplementary Table 2). We are a large cord blood transplant facility; there are a higher number of UCBT patients than PBCST patients. Our retrospective study is a real long-term retrospective study. We evaluated outcomes in patients who had high-risk or recurrent refractory hematological malignancies and underwent transplantation between April 2001 and June 2017. Patients were eligible if they underwent a myeloablative conditioning regimen and received a single unrelated cord blood unit or allogeneic peripheral blood stem cells.

Supplementary Table 2. Demographic and clinical characteristics of patients in the retrospective study.

	UCBT patients (N=439)	PBSCT patients (N=162)
Median age (range), yr.	12 (1-70)	32 (1-62)
Median weight (range), kg	40 (8-100)	60 (14-97)
Male sex, n (%)	279 (63.6)	103 (63.6)

Sex (donor/recipient), n (%)		
Male/male	137 (31.2)	55 (34.0)
Male/female	85 (19.4)	37 (22.8)
Female/male	140 (31.9)	47 (29.0)
Female/female	73 (16.6)	22 (13.6)
Missing data	4 (0.9)	1 (0.6)
Diagnosis, n (%)		
Acute myeloid leukemia	156 (35.5)	59 (36.4)
Acute lymphoblastic leukemia	219 (49.9)	54 (33.3)
Myelodysplastic syndrome	27 (6.2)	18 (11.1)
Mixed lineage leukemia	4 (0.9)	2 (1.2)
Chronic myeloid leukemia	27 (6.2)	26 (16.0)
Lymphoma	6 (1.4)	2 (1.2)
Multiple myeloma	0 (0)	1 (0.6)
Conditioning regimen, n (%)		
TBI+CY+/-others	150 (34.2)	47 (29.0)
BU+CY+/-others	289 (65.8)	115 (71.0)
GVHD prophylaxis, n (%)		
CsA+MMF	433 (98.6)	114 (70.4)
CsA+MMF+MTX	6 (1.4)	34 (21.0)
CsA+MMF+ATG	0 (0)	4 (2.5)
CsA+MTX	0 (0)	3 (1.9)

CsA+MMF+CY	0 (0)	3 (1.9)
CsA+ATG	0 (0)	1 (0.6)
CsA+MMF+ATG+MTX	0 (0)	3 (1.9)

Abbreviations: TBI, total body irradiation; CY, cyclophosphamide; BU, busulfan; GVHD, graft-versus-host disease; CsA, cyclosporine A; MMF, mycophenolate mofetil; MTX, methotrexate; ATG, anti-thymocyte globulin.

2. Similarly, there were no baseline information about patients with and without PES and PES patients and whether adjusting for confounding is required.

Author response: We apologize for causing confusion. We collected baseline information for UCBT patients with and without PES (as shown in Supplementary Table 3 below), and we found little statistical difference.

Supplementary Table 3. Demographic and clinical characteristics of UCBT patients with and without PES.

	PES (N=335)	Non-PES (N=104)	p
Median age (range), yr.	12 (1-64)	14 (1-70)	
Median weight (range), kg	40 (9-82)	45 (8-100)	
Male sex, n (%)	200 (59.7)	79 (76.0)	
Diagnosis, n (%)			0.185
Acute myeloid leukemia	116 (34.6)	40 (38.5)	
Acute lymphoblastic leukemia	175 (52.2)	44 (42.3)	

Myelodysplastic syndrome	18 (5.4)	9 (8.7)	
Mixed lineage leukemia	4 (1.2)	0 (0)	
Chronic myeloid leukemia	19 (5.7)	8 (7.7)	
Lymphoma	3 (0.9)	3 (2.9)	
Conditioning regimen, n (%)			0.912
TBI+CY+/-others	114 (34)	36 (34.6)	
BU+CY+/-others	221 (66)	68 (65.4)	
GVHD prophylaxis, n (%)			0.148
CsA+MMF	332 (99.1)	101 (97.1)	
CsA+MMF+MTX	3 (0.9)	3 (2.9)	

Abbreviations: GVHD, graft-versus-host disease; TBI, total body irradiation; CY, cyclophosphamide; BU, busulfan; CsA, cyclosporine A; MMF, mycophenolate mofetil; MTX, methotrexate.

3. Figure 2a to Figure 2h – number included in the analysis were small and would like to know if it was because due to data availability or were randomly selected. A lot of analyses carried out were performed (including multiple outcomes and multiple time point analyses) and would be good to indicate if type I error were accounted for in these comparisons, and justify if not.

Author response: We apologize for causing confusion. In the mechanistic studies, to study the dynamic changes for the cytokine and cellular detection in Figure 2a to Figure 2h, we acquired blood samples **on a daily basis** within two weeks after UCBT. Due to ethical limitations, it was difficult to carry out experiments with a larger sample size. The small number of samples included in the analysis is due to sample limitations rather than selection bias.

4. I am probably not too worried about using t-test for small sample data but I think some of the data reported in this manuscript were skewed.

Author response: Thank you! This is a real study, but the sample size was small due to ethical limitations. All the sample data were tested for a normal distribution with the Kolmogorov-Smirnov test. The *t*-test or one-way ANOVA was performed for data with a normal distribution (Figure 2h, Supplementary Figure 5c, and Supplementary Figure 7a), and a non-parametric test was performed for data with a non-normal distribution (Figure 2a, Figure 3g, Figure 4e, Supplementary Figure 3, Supplementary Figure 4, Supplementary Figure 5b, and Supplementary Figure 7b). We appreciate your understanding and support.

5. If comparisons between group were made, then if possible, the difference between the group and corresponding 95% CI should be reported rather than by group results.

Author response: Thank you. We appreciate your insightful suggestion. In the revised manuscript, for the normal distribution data, we have reported the corresponding 95% CI according to your suggestion (Figure 2h, Supplementary Figure 5c, and Supplementary Figure 7a). Since our data are real clinical data, some have a non-normal distribution. Thank you very much for your understanding and support.

Figure 2h. Donor chimerism in the peripheral blood of recipients on day 7 post-UCBT. One-way ANOVA (Non-PES vs. PES-0: mean difference=-11.46, 95% CI: -26.42 to 3.499; Non-PES vs. PES-1: mean difference=-38.79, 95% CI: -55.56 to -22.02; Non-PES vs. PES-2: mean difference=-44.34, 95% CI: -61.89 to -26.80). Each symbol represents an independent individual. Data are shown as mean \pm standard error of the

mean (SEM). n.s., not significant, and **** $p < 0.0001$.

Supplementary Figure 5c. ELISA of GM-CSF in supernatants from peripheral blood stem cells ($n = 6$) or cord blood mononuclear cells ($n=11$) cultured for 6 hours. Mean difference= 23.99 , 95% CI: $2.158-45.82$, p (t -test) = 0.034 . PBSC, peripheral blood stem cells. CB, cord blood mononuclear cells. Each data point represents a biologically-independent sample. Data are presented as mean \pm SEM. * $p < 0.05$.

Supplementary Figure 7a. Plasma IL-6 levels were significantly higher in patients with PES than those without PES. Mean difference= -41.42 , 95% CI: -56.08 to -26.76 , p (t -test) < 0.0001 . Data are presented as mean \pm SEM. **** $p < 0.0001$.

6. The single arm study was reported briefly in the manuscript. There was no justification on sample size and the rationale of why this study did not have a control group but was compared with historical controls from another observational study, which no information was provided about this study. Again, no baseline information were reported and therefore difficult to access the comparability between the two groups. There were 3 PES subgroups who received different level of intervention in the observational study and it isn't clear which PES subgroup was use as the control group

and why. I suggest the authors should follow the CONSORT statement as far as possible if the authors felt this was a clinical trial.

Author response: We apologize for causing confusion. At present, steroid treatment is the first-line therapy for PES. Symptoms improve promptly in patients with mild PES following the initiation of intravenous systemic corticosteroid therapy, while patients with severe PES had no response even when the dosage of methylprednisolone was up to 2 mg/kg/day. Retrospective studies have shown a high mortality rate in patients with severe PES (methylprednisolone, 2 mg/kg/day). In mechanistic studies, we found that IL-6 may be the cause of severe PES. To verify whether targeting IL-6 signals may potentially save patients with severe PES, we applied for a clinical trial using tocilizumab to block the IL-6 receptor, and we used the historical 2 mg/kg/day methylprednisolone subgroup as the control group for ethical considerations. The baseline information of patients with severe PES for the single-arm study and the historical controls' study is shown below (Supplementary Table 4).

Supplementary Table 4. Demographic and clinical characteristics of patients with severe PES in the single-arm study and the historical controls' study.

	The single-arm study (N=11)	The historical controls' study (N=50)	p
Median age (range), yr.	5 (2-22)	10 (1.5-50)	
Median weight (range), kg	19 (10-43)	32 (9-82)	
Male sex, n (%)	6 (54.5)	26 (52.0)	
Sex (donor/recipient),			0.188
Male/male	4 (36.4)	14 (28)	
Male/female	1 (9.1)	17 (34)	

Female/male	2 (18.2)	12 (24)	
Female/female	4 (36.4)	7 (14)	
Diagnosis, n (%)			0.27
Acute myeloid leukemia	4 (36.4)	14 (28)	
Acute lymphoblastic leukemia	4 (36.4)	30 (60)	
Myelodysplastic syndrome	2 (18.2)	4 (8)	
Chronic myeloid leukemia	1 (9.1)	1 (2)	
Lymphoma	0 (0)	1 (2)	
Conditioning regimen, n (%)			1
TBI+CY+/-others	2 (18.2)	8 (16)	
BU+CY+/-others	9 (81.8)	42 (84)	
GVHD prophylaxis, n (%)			0.18
CsA+MMF	10 (90.9)	50 (100)	
CsA+MMF+MTX	1 (9.1)	0 (0)	

Abbreviations: TBI, total body irradiation; CY, cyclophosphamide; BU, busulfan; GVHD, graft-versus-host disease; CsA, cyclosporine A; MMF, mycophenolate mofetil; MTX, methotrexate.

7. In the Discussion section, the authors have stated “We successfully showed that tocilizumab can effectively improve signs and symptoms of PES”. I find this statement is too strong and the treatment recommendation is worryingly flawed. based on findings from a very small single arm study and compared with historical "controls".

Author response: Thank you. According to your suggestion, the sentence has been revised (**page 12, line 277**). The revised sentence reads, "These clinical results suggest

that tocilizumab may effectively improve signs and symptoms of PES."

Reviewers' comments:

Reviewer #3 (Remarks to the Author):

The authors have addressed most of the comments I raised. One item that was not addressed was about the controlling for type I error due to multiple testing. I hope the author can make this clear/or address in the manuscript regarding this issue.

Reviewer #4 (Remarks to the Author):

General comment

Jin et al. report the results of 1) a large retrospective study assessing the frequency of pre-engraftment syndrome (PES) in patients given UCBT versus PBSCT ; 2) mechanistic studies comparing PB and CB monocytes and their ability to produce GM-CSF leading to IL-6 production; 3) a small prospective study assessing the administration of tocilizumab in patients with steroid-refractory PES. Although the findings are of interest i have some concerns regarding the methodology.

Specific comments.

Major comments

1) Line 35. The authors did not demonstrate that tocilizumab administration improved OS. An appropriate randomized study would be necessary to establish this hypothesis (and not a comparison of data from 11 prospective patients versus historical controls).

2) Line 113. It is unclear how the gradation scale for PES was developed. This should have been done in a training set and validated in a validation set.

3) Lines 222 - 232. I have some questions regarding this prospective study. First, it is unclear what was the phase of the study and how the number of patients to be included was selected. In the website link provided by the author (www.chictr.org.cn (Reference: ChiCTR1800015472)) the study is reported to be a phase IV study. I would have expected this study to be either a phase I or a phase I-II study. For a phase I study, the primary endpoints reported (fever resolution and 6-months TRM) are not appropriate. For a phase II study, the statistical power should have been calculated upfront (according to the study hypothesis) and the number of included patients should have been defined accordingly. I have also some concerns regarding the number of patients included since in the website the study was designed to include 10 patients while data from 11 patients are reported in the manuscript.

Minor comments

1) Line 49. This is not true that it is possible to find a sufficiently rich cord blood unit for all patients. This is especially not the case in patients with a high body weight.

2) Line 52. In europe cord blood transplantation is dramatically decreasing in contrast to HLA-haploidentical transplantation. Further, 2 recent randomized studies have demonstrated better outcomes with HLA-haploidentical transplantation in comparison to cord blood transplantation (doi: 10.1182/blood.2020007535 and doi: 10.1016/j.bbmt.2019.10.014). Also higher GVL effect with CB has not been identified in these randomized studies neither than in a large EBMT retrospective study (doi: 10.1158/1078-0432.CCR-17-3622). This should be discussed.

3) Administering tocilizumab after UCBT is already assessed in a large phase II study in US (<https://clinicaltrials.gov/ct2/show/NCT03434730>). This should be discussed.

REVIEWER COMMENTS

Reviewer #3 (Remarks to the Author):

The authors have addressed most of the comments I raised. One item that was not addressed was about the controlling for type I error due to multiple testing. I hope the author can make this clear/or address in the manuscript regarding this issue.

Author response: Thank you! We appreciate your comments. For multiple testing, we refer to the significance level α of 0.01 to reduce type I error. In the revised manuscript, we have made this clear in Methods (**page 19, line 410-411**). Thank you very much for your understanding and support.

Reviewer #4 (Remarks to the Author):

General comment

Jin et al. report the results of 1) a large retrospective study assessing the frequency of pre-engraftment syndrome (PES) in patients given UCBT versus PBSCT; 2) mechanistic studies comparing PB and CB monocytes and their ability to produce GM-CSF leading to IL-6 production; 3) a small prospective study assessing the administration of tocilizumab in patients with steroid-refractory PES. Although the findings are of interest I have some concerns regarding the methodology.

Author response: Thank you. We appreciate your comments and suggestions.

Specific comments.

Major comments

1) Line 35. The authors did not demonstrate that tocilizumab administration improved OS. An appropriate randomized study would be necessary to establish this hypothesis (and not a comparison of data from 11 prospective patients versus historical controls).

Author response: We apologize for this oversight. “OS” has been replaced by “non-relapse mortality” in the revised manuscript (**page 2, line 36**). This study identified the pathogenesis of PES and found that cord blood-derived monocytes have inflammatory phenotypes, they play a vital role in driving PES by producing GM-CSF and IL-6.

Finally, we tried to demonstrate this mechanism in a preliminary clinical study. At present, steroid treatment is the first-line therapy for PES, while patients with severe PES had no response even when the dosage of methylprednisolone was up to 2 mg/kg/day. Retrospective studies have shown a high mortality rate in patients with severe PES (methylprednisolone, 2 mg/kg/day). To verify whether targeting IL-6 signals may potentially save patients with severe PES, we applied for a clinical trial using tocilizumab to block the IL-6 receptor, and we used the historical 2 mg/kg/day methylprednisolone subgroup as the control group for ethical considerations. We hope to get your appreciation and support.

2) Line 113. It is unclear how the gradation scale for PES was developed. This should have been done in a training set and validated in a validation set.

Author response: We apologize for causing confusion. According to a retrospective study, we found that symptoms occur within 7 days after UCBT, more than two symptoms (rash, diarrhea, abdominal pain, hypoxia, cough, edema), non-responsive to corticosteroids are independent risk factors for the severity of PES patients, which were then validated in a study of the risk classification and stratified intervention on PES after UCBT (Reference: ChiCTR-ONC-16009013). We developed the gradation scale for PES based on the number of risk factors (as shown in Supplementary Table 1 below).

Supplementary Table 1: Grading system used for pre-engraftment syndrome (PES).

Risk factors	
1. Symptoms occur < 7 days after UCBT*	
2. More than two symptoms (rash, diarrhea, abdominal pain, hypoxia, cough, edema)	
3. Non-responsive to corticosteroids	
Grading system	
Grade 0	Lack of a risk factor; symptoms are minor and mild; responds quickly to corticosteroids
Grade 1	Presence of one risk factor
Grade 2	Presence of two risk factors
Grade 3	Presence of three risk factors
* UCBT denotes unrelated cord blood transplantation.	

3) Lines 222 - 232. I have some questions regarding this prospective study. First, it is unclear what was the phase of the study and how the number of patients to be included was selected. In the website link provided by the author (www.chictr.org.cn (Reference: ChiCTR1800015472)) the study is reported to be a phase IV study. I would have expected this study to be either a phase I or a phase I-II study. For a phase I study, the primary endpoints reported (fever resolution and 6-months TRM) are not appropriate. For a phase II study, the statistical power should have been calculated upfront (according to the study hypothesis) and the number of included patients should have been defined accordingly. I have also some concerns regarding the number of patients included since in the website the study was designed to include 10 patients while data from 11 patients are reported in the manuscript.

Author response: We apologize for causing confusion. Our study is an investigator-initiated proof of concept study to verify the mechanism by which IL-6 produced by cord blood-derived inflammatory monocytes mediate PES. In order to standardize the study, we conducted this prospective study. We noted that clinical trials of extended indications for post-marketing drugs are classified as phase II studies by the FDA. In

China, the National Medical Products Administration currently has no unified standard for the phase of investigator-initiated proof of concept study.

Considering that some patients might be lost to follow-up, and the retrospective study have shown a high mortality rate in patients with severe PES, patients with severe PES who had no response even when the dosage of methylprednisolone was up to 2 mg/kg/day volunteered to be enrolled. We ended up enrolling 11 patients. Thank you very much for your understanding and support.

Minor comments

1) Line 49. This is not true that it is possible to find a sufficiently rich cord blood unit for all patients. This is especially not the case in patients with a high body weight.

Author response: Yes, we agree with you. We apologize for this oversight. Comprehensive consideration, the sentence has been revised (**page 2, line 43-46**). The revised sentence reads, "HLA-haploidentical transplantation has spread rapidly worldwide, and cord blood (CB) is also a good alternative source of hematopoietic stem cells. CB has many advantages as a stem cell source. For example, CB is more permissive of HLA disparity due to its lower immunogenicity."

2) Line 52. In Europe cord blood transplantation is dramatically decreasing in contrast to HLA-haploidentical transplantation. Further, 2 recent randomized studies have demonstrated better outcomes with HLA-haploidentical transplantation in comparison to cord blood transplantation (doi: 10.1182/blood.2020007535 and doi: 10.1016/j.bbmt.2019.10.014). Also higher GVL effect with CB has not been identified in these randomized studies neither than in a large EBMT retrospective study (doi: 10.1158/1078-0432.CCR-17-3622). This should be discussed.

Author response: We appreciate your important suggestion. We have revised the sentence (**Page 3, Line 49-50**). According to your suggestion, we have now cited these literatures and provided additional discussion relating to this point (**Page 11-12, Line 254-259**). The sentences are "Recent studies have reported discordant results, HLA-haploidentical stem cell transplantation with post-transplantation cyclophosphamide

provides improved outcomes compared with ATG-containing UCBT³⁶⁻³⁸. Some studies have found a deleterious effect of exposure of ATG in UCBT, results might be improved by omitting ATG from the conditioning regimen^{39,40}. Further comparisons are warranted to better donor selection".

References

(The numbers given after each of the cited references refer to those in the reference list of our manuscript, and the cited references are listed below.)

36 Fuchs, E. J. et al. Double unrelated umbilical cord blood vs HLA-haploidentical bone marrow transplantation: the BMT CTN 1101 trial. *Blood* 137, 420-428, doi:10.1182/blood.2020007535 (2021).

37 Sanz, J. et al. Prospective Randomized Study Comparing Myeloablative Unrelated Umbilical Cord Blood Transplantation versus HLA-Haploidentical Related Stem Cell Transplantation for Adults with Hematologic Malignancies. *Biol Blood Marrow Transplant* 26, 358-366, doi:10.1016/j.bbmt.2019.10.014 (2020).

38 Baron, F. et al. Impact of Donor Type in Patients with AML Given Allogeneic Hematopoietic Cell Transplantation After Low-Dose TBI-Based Regimen. *Clin Cancer Res* 24, 2794-2803, doi:10.1158/1078-0432.Ccr-17-3622 (2018).

39 Pascal, L. et al. Impact of rabbit ATG-containing myeloablative conditioning regimens on the outcome of patients undergoing unrelated single-unit cord blood transplantation for hematological malignancies. *Bone Marrow Transplant* 50, 45-50, doi:10.1038/bmt.2014.216 (2015).

40 Zheng, C. et al. Comparison of conditioning regimens with or without antithymocyte globulin for unrelated cord blood transplantation in children with high-risk or advanced hematological malignancies. *Biol Blood Marrow Transplant* 21, 707-712, doi:10.1016/j.bbmt.2014.12.023 (2015).

3) Administering tocilizumab after UCBT is already assessed in a large phase II study in US (<https://clinicaltrials.gov/ct2/show/NCT03434730>). This should be discussed.

Author response: Thank you. We appreciate this important point from the reviewer and have provided additional discussion relating to this point (**Page 13, Line 280-283**).

The sentences are "Coincidentally, we noticed that a study sponsored by Memorial Sloan Kettering Cancer Center using tocilizumab to ameliorate aGVHD and early toxicity after double UCBT is ongoing (<https://clinicaltrials.gov> (NCT03434730)). We will pay close attention to the results of the study".

Reviewers' comments:

Reviewer #4 (Remarks to the Author):

The authors have addressed the comments I raised.